# Experiences of integrating a psychological intervention into a youth-led empowerment program targeting out-of-school adolescents, in urban informal settlements in Kenya: A qualitative study

**Beth Kangwana**[1]☯*, **Joan Mutahi**[2]☯, **Manasi Kumar**[2]☯

**1** Population Council-Kenya, Nairobi, Kenya, **2** Institute for Excellence in Health Equity, New York University Grossman School of Medicine, NY, NY, United States of America

☯ These authors contributed equally to this work.
* bkangwana@popcouncil.org

**Data Availability Statement:** The data underlying the results presented in the study are available

## Abstract

### Introduction

Depression, anxiety and behavioural disorders are the leading causes of illness and disability in adolescents. This study aims to evaluate the feasibility of integrating mental health services into a youth-led community-based intervention targeting out-of-school adolescents, residing in Kariobangi and Rhonda informal settlements in Kenya.

### Method

Youth mentors were trained on the Bridging the Gaps (BTG) curriculum that integrated a modified version of the World Health Organization's (WHO) Problem Management Plus (PM+) psychological intervention into a sexual health, life-skills and financial education curriculum. Community lay mentors facilitated 72 weekly group sessions for 469 adolescent boys and girls, augmented with five enhanced one-on-one treatment sessions for those displaying signs of psychological distress. Adolescents displaying severe signs of psychological distress were referred directly to a primary health facility or connected to specialist services. A qualitative survey took place between February and March 2022, around four months before the end of the program. In-depth interviews were carried out with 44 adolescents, 7 partners, 19 parents and 11 stakeholders. Four focus group discussions were carried out with 17 mentors. Respondents were purposively selected to be interviewed based on their level of exposure to the intervention and ability to provide in-depth experiences. Themes focused on the program's perceived effectiveness, ability to develop the capacity of lay mentors to address mental health issues, and increased access to mental health services.

### Results

Adolescents reported that the intervention was able to improve their confidence in speaking up about their problems, equip them with essential first-aid skills to manage and treat anxiety

through: Kangwana, Beth, 2023, "Experiences of integrating a psychological intervention into a youth-led empowerment program", https://doi.org/10.7910/DVN/W8NWWD, Harvard Dataverse, V1.

**Funding:** The funders had no role in study design, data collection and analysis, decision to publish, or preparation of the manuscript.

**Competing interests:** The authors have declared that no competing interests exist.

or mild depression, provide them access to free one-on-one psychological help sessions, and increase their social network. Mentors were able to adhere to the core principles of psychological intervention delivery, providing preventative and treatment-focused psychosocial services. Furthermore, parents reported experiencing improved adolescent receptivity to parental suggestions or advice leading to improved parent-adolescent relationships. Mentors referred adolescents for a variety of reasons including severe mental illness, rape, and alcohol and substance use however, the high cost of transport was the main barrier limiting adolescents from following through with their referrals.

## Conclusion

The findings demonstrate that integration of mental health services into community-based interventions is feasible and has benefits for adolescents, parents, and mentors.

## Introduction

Globally, approximately one in four adolescents experience a mental illness, with depression, anxiety and behavioural disorders being leading causes of illness and disability among this age group [1–3]. According to the Kenya Adolescent Mental Health Survey, up to 26% of adolescents 10–17 years suffer from anxiety [4]. Out-of-school adolescents living in urban informal settlements are more likely to be exposed to adverse social determinants of mental health compared to their counterparts residing outside of these settings and are therefore significantly more likely to develop a mental illness [1, 3, 5]. These determinants include a lack of economic security, unequal gender norms, pressure from peers, childhood adversity, violence, not being in employment or education, and exposure to social media [1, 3, 5]. Adolescents with mental health conditions are more likely to suffer from stigmatization, isolation, poor help-seeking behaviours, risk-taking behaviours, violence, poor sexual and reproductive health (SRH) and human rights violations [1, 3, 5].

In lower- and middle-income countries, there remains a gap between the burden of mental health disorders and availability of services, where only one out of 27 people with depression receive adequate services compared to one out of five in upper middle-income countries [6]. In order to mitigate barriers to access, recommendations have been made to scale up mental health interventions for early-stage screening, especially for low intensity problems, through training of lay and non-specialist health workers, also known as task sharing and task-shifting [1]. One example is the Problem Management Plus (PM+) program, developed by the World Health Organisation (WHO) in 2016, as a low-intensity five-session intervention to address symptoms of common mental health problems such as depression, anxiety, stress or grief in adults residing in low resource communities [7]. The intervention is intended to be delivered by trained community lay workers and has demonstrated effectiveness in reducing psychological distress and depression symptoms through strengthening psychosocial skills and implementing behavioral strategies of stress management in day-to-day life [7, 8].

Both global and national Kenyan policies recommend the integration of mental health services into other community-based services to provide a more holistic approach to addressing the multitude of preventative and curative health needs of adolescents [1, 9]. This is because, aside from mental illness, the critical stage of cognitive and social transition adolescents undergo, combined with the adverse social determinants mentioned above, increases their risks of experiencing an array of negative health outcomes such as unintended early

pregnancies, and exposure to sexually transmitted infections (STIs), including HIV [5]. Despite this recommendation, there have been few studies in sub–Saharan Africa (SSA) that evaluate the integration of mental health services into community-based health interventions. A systematic review carried out by Musindo *et al.*, (2023) identifying whether and how interventions targeting adolescent SRHR and HIV in SSA include mental health components showed that out of 27 studies, mental health concerns were only addressed in two [10].

The general objective of the study is therefore to determine the feasibility of embedding the WHO's PM+ mental health program into an evidence-based social, health and economic community empowerment-building program for out-of-school adolescents. The specific objectives are to understand the impact of the integration on:

1. Perceived changes in psychosocial outcomes for vulnerable adolescents;

2. The capacity of community youth mentors to address mental health challenges of vulnerable adolescents, and a deliver multilayered intervention for vulnerable youth;

3. Vulnerable adolescents' access to quality mental health information and services in the community.

## Methods

### The intervention

Nineteen lay mentors were trained on the Bridging the Gaps (BTG) curriculum (see further details below) for a period of three weeks. Some of the training sessions were carried out online because they occurred during the initial months of the COVID pandemic, when the Kenyan government's mitigation guidelines prohibited large gatherings. The mean age of the mentors was 29 years (range 23 to 35 years) of which six were males and the rest females. The BTG program consisted of group sessions led by the trained youth mentors who resided in the targeted communities. The sessions were carried out once a week for a period of 18 months, and each session lasted around 90 minutes. During the sessions, mentors facilitated the delivery of the curriculum. Separate group sessions were held for boys and girls for a series of facilitated sessions however five months into the sessions, one meeting per month was joint between boys' and girls' groups, which was co-facilitated by a female mentor and a male mentor. The joint sessions were used to build awareness of the experiences of gender and to practice gender-sensitive behaviors, and were followed by a debrief back in the gender-separated groups. During the group sessions, enhanced one-on-one treatment sessions were offered by mentors to adolescents who were displaying signs and symptoms of significant psychological distress (defined as heightened scores on Patient Health Questionnaire (PHQ)-9 [11], and a positive score on item 9 (indicating suicidality) during scheduled screening timepoints of the study). Sessions were also offered to adolescents who at any time during the intervention explicitly mentioned or were intuitively identified by the mentors as feeling disturbed. Adolescents displaying signs of severe mental disturbance were referred directly to a primary healthcare facility or connected to specialist services. As part of the referral process mentors would let the adolescents understand the reason for the referral, the place where they were being referred, and would provide a referral note to the mentee to hand over to the referral facility on arrival.

The BTG program was implemented from January 2021 to July 2022. Adolescents were recruited through door-to-door recruitment by mentors from two community-led organisations (COs). During the recruitment, potential participants and their parents were informed about the study and asked if they were willing to participate. Mentors used the snowballing technique where they asked households they visited if they knew of other adolescents nearby

who they felt would be interested in participating. Adolescents were eligible if they were 1) out of school, 2) 15–18 years of age and 3) usually residing in the study area. Following the door-to-door recruitment, adolescents, their parents, guardians or caregivers and partners were invited to an enrollment activity that was held in a nearby community facility where they were provided with more information about the program. During this time baseline data collection activities took place on the characteristics of the adolescent study participants. A total of 469 adolescents were recruited into the study of which around two-thirds (292) were female and a negligible number (<1%) were categorized under "other". It was intentional to recruit a larger proportion of females because they tend to be more marginalized in these settings compared to males. In addition, the study group had more experience carrying out studies on females and was still learning how to integrate males into female-centered programming. There were no restrictions on the number of adolescents who could be selected from any one household.

## The curriculum

The BTG curriculum embeds a mental health component into sexual and reproductive health and rights, life skills and financial education (SRHLS) topics. A large component of the SRHLS curriculum for girls' was derived from the Adolescent Girls' Initiative- Kenya curriculum [12] developed by the Population Council, slightly adapted to factor in the study participants' age and context. The male curriculum was derived from Tuko Pamoja, a curriculum developed by the Population Council in collaboration with PATH [13], and UNEP's Engaging Men as Partners [14]. The SRHLS curriculum has been previously evaluated through randomized controlled trials and has shown to significantly increase adolescents' sexual and reproductive health knowledge, condom self-efficacy, savings behavior, positive gender norms and attitudes, and self-efficacy [12, 15, 16].

The mental health component of the intervention was derived from the WHO PM+ program. This is a 5-session individual psychological intervention that can be delivered by non-specialist counsellors and addresses common mental disorders in people affected by adversity [7, 8, 17]. WHO more recently developed PM+ to be delivered in group sessions however, this manual was released towards the end of this study. PM+ in its original format was meant for adults experiencing symptoms of common mental health problems including depression, anxiety, stress or grief, as well as self-identified practical problems such as unemployment and interpersonal conflict [7]. In the BTG adaptation, it was infused with youth responsive stances, self-care, psychological first-aid informed ideas, and core competencies to work with young people in Kenyan informal settlements. In addition, a mapping exercise was undertaken to build a referral pathway to formal services for those who required more professional healthcare interventions. The curriculum for each topic was delivered during separate sessions however, mentors were encouraged by the supervisors to take the initiative of interweaving the topics learnt from the mental health curriculum into the other sessions with the aid of case studies. The curriculum was translated into Kiswahili, which is predominantly spoken in the urban informal settlements and is one of Kenya's official languages.

The BTG intervention was co-created by a group of clinical psychologists, two community-led organizations (COs) and the Population Council, a research-based institution. Population Council-Kenya oversaw the program and implemented the monitoring and evaluation activities. BTG was implemented by the COs: Integrated Education for Community Empowerment (IECE) in Kariobangi, Nairobi and REPACTED in Rhonda, Nakuru. The COs were responsible for the implementation of the intervention and had previous extensive experience running empowerment building group sessions within their respective communities, including implementation of the SRHLS curriculum component [18]. Further details of the intervention design can be found in the team's process paper [19].

## Study site

The study was carried out in two urban informal settlements, Kariobangi in Nairobi and Rhonda in Nakuru. Adolescents living in such settlements are generally exposed to higher levels of negative social adversities such as limited access to education and information (including SRHR and mental health), and minimal government services [20]. Studies in informal settlements have shown that caregivers' limited knowledge, embarrassment, and cultural taboos prevent them from providing adolescents with accurate information on sexual and reproductive health [21].

Adolescent characteristics from the baseline sample showed that one-quarter were diagnosed with moderate to severe depression with a mean score of 6.3 and 7 out of 27 for males and females respectively; this was measured using the PHQ-9 [11]. Data on alcohol and substance use, measured using the WHO Alcohol Use Disorders Identification Test (AUDIT) tool [22] showed that just under half (45%) of males and 19% of females required some intervention for cannabis use. Overall, participants scored above average on the mental health literacy score (61%) [23] and reported being most likely to seek help from or report suicidal ideation to a parent, a doctor including a mental health professional or a mentor. A total of 72% of adolescents reported ever having sex, of which 24% were male and 31% were females. Of all the females, just under 40% reported ever giving birth and a tenth reported currently being pregnant. Just under a half (42.1%) of all adolescents reported engaging in transactional sex, and 15% reported suffering from some form of disability.

## Qualitative study design and data collection

The qualitative survey was nested within a pre-post intervention study. Qualitative interviews took place between February and March 2022, around four months before the end of the program.

In-depth interviews (IDIs) were carried out with 44 adolescents, 7 partners, 19 parents and 11 stakeholders. Four focus group discussions (FGDs) were carried out with 17 mentors (Table 1). The number of interviews was determined by what could be feasibly achieved given the budget and available time however, by the end of the interviews data saturation seemed to have been achieved [24]. The topics in the semi-structured interview guides covered general knowledge and awareness of the program, experiences of the mentor training sessions, safe space meetings, one-on-one sessions, mentor capabilities, and referral processes to formal healthcare services.

**Table 1. Number of in-depth interviews (IDIs) and focus group discussions (FGDs) conducted by site and respondent type.**

| Respondent Type | | IDIs | | FGDs | | Totals |
|---|---|---|---|---|---|---|
| | | Kariobangi | Rhonda | Kariobangi | Rhonda | |
| **Mentors** | | | | 2[#] (10) * | 2[#] (7) * | 4 |
| **Adolescents** | Boys | 13 | 9 | | | 22 |
| | Girls | 11 | 11 | | | 22 |
| **Parents** | | 7 | 12 | | | 19 |
| **Partners** | | 5 | 2 | | | 7 |
| **Stakeholders**[+] | | 6 | 5 | | | 11 |
| **Total** | | 42 | 39 | 2 | 2 | 85 |

* Total number of individuals interviewed through the FGDs

[#] Total number of FGDs

[+] Stakeholders included religious leaders, teachers, community members in the legal profession and Community Health Volunteers (CHVs)

For each of the sites, adolescents who had higher rates of attendance were purposively selected for the IDIs because they had the greatest exposure to the program and were able to provide more in-depth experiences regarding the program (S1 Table). Parents of the adolescents interviewed were also purposively selected based on their knowledge of the program, ability to make the time and willingness to be interviewed. Community stakeholders were purposively selected based on their awareness of, and involvement with the BTG program (S1 Table). Respondents were approached in-person by the mentors and were invited to participate in the interviews. Those who were not able to participate were replaced with other respondents, however, participation was possible for all respondent categories apart from the fathers where we were unable to find one replacement within the interview time period. The main reason provided for non-participation was competing work or home-related activities.

Interviews were carried out using semi-structured interview guides that provided the main questions and probes to elicit in-depth responses from respondents. The guides were developed to address the objectives of the qualitative survey and explore opinions and experiences of each component of the intervention. They were primarily designed by the research team and reviewed by the program's clinical psychologists and CO implementors from both sites. The guides were translated from English to Kiswahili by a consultant translator, and the translations were reviewed by the data collection team who were also conversant in English and Kiswahili. The guides were pre-tested in the study sites with non-study participants and where necessary, revised to ensure questions were being delivered and interpreted appropriately. All interviews were recorded using digital tape recorders and interviewers were provided with note pads to document any observations that would likely enrich the interpretation of the interviews, such as non-verbal communication. Notes were taken both during and after the interviews. At the time of the interviews, a brief sociodemographic form was completed for each IDI and FGD respondent (S1 Table). The form captured information on the study site, interview location, and respondent characteristics such as age, gender, education level, marital status, religion and occupation. The IDIs averaged one hour and 30 minutes while the FGDs went on for around three hours, and no repeat interviews were carried out. Interviews were carried out in hired-out private rooms in schools and churches within the study sites. Only respondents and interviewers were present in the rooms during the interviews.

Interviews were carried out by male and female interviewers, employed as research assistants, who all had university degrees in social science and had previous experience conducting qualitative research. Prior to carrying out the interviews, interviewers underwent a one-week training on research ethics and interviewing techniques. Due to the sensitivity of some of the questions and to allow adolescents to open up, interviewers interviewed adolescents who were of the same gender. In order to build rapport with the respondent, interviewers would introduce themselves, including their name and organization they work for, to the respondent, and after obtaining consent respondents were free to ask interviewers any questions they had regarding the study or the interviewer. Each interview would then commence with a general chat about the BTG program.

## Analysis

All IDIs and FGDs were recorded with participant permission and transcribed directly into English. Transcripts were validated and reviewed for quality assurance by a second validator prior to being coded; they were not returned to the participant for further amendments. No personal identifying information, other than the assigned participant identification numbers, were included in the transcriptions. Transcripts were organized according to data collection method (FGD vs. IDI), study site, and respondent type, and uploaded into the research software NVivo (Version 12) for coding.

A combination of content analysis and grounded theory was used to create the study codes, which arose both through reviewing the structure of the interview guides and identifying the themes that emerged from the transcripts themselves [25, 26]. An initial starting list of codes was developed and included in a code-book based on the program's theory of change and interview guides, and then additional codes were added as new themes emerged from the data [27] (S2 Table). In the end, codes were categorized into general knowledge and perceptions of the overall program, mentor training sessions, safe space group meetings, one-on-one sessions and the referral process. All transcripts were double coded by two qualified analysts using NVivo 12. To test for intercoder agreement across the double-coded transcripts, a Krippendorff's c-α-binary coefficient was obtained for all key codes. For cases where the coefficient was less than 0.70, side-by-side comparison, clarification, and reconciliation were carried out on the specific coded transcripts. A coding matrix was then generated by code and sub-code and this was used to interpret the findings. Mentors and mentees were provided with a summary of the findings during a dissemination meeting where they were able to provide feedback; no transcripts were returned to the participants for comments.

## Ethical considerations

The study protocol was approved by the Population Council Institutional Review Board (Protocol 946) and the African Medical and Research Foundation (AMREF) Ethical and Scientific Review Committee (P916-2020). In addition, the protocol was reviewed by the Kenyan National Commission for Science, Technology and Innovation to obtain research permits for study investigators (NACOSTI/P/21/8948). A couple of weeks prior to data collection, the CO staff approached the parents of potential adolescents, below 18 years, to be interviewed, and obtained written consent for their adolescent to participate in the qualitative study. At the beginning of any interview, written consent was obtained from respondents 18 years and above as well as emancipated minors. Written assent was obtained from adolescents below 18 years. Prior to joining the program, both mentees and mentors signed a confidentiality agreement form; mentors also underwent a safeguarding training on child protection.

## Results

### Background characteristics of respondents

A total of 44 adolescent boys and girls were interviewed, half of whom were female (S1 Table). The mean age was 18 years. Of those interviewed, 41% of girls and 55% of boys had incomplete secondary education. Around 84% of adolescents had never been married and a small minority (9%) were currently married. A large majority (81%) of adolescents were Christian with around 30% being Catholic and the rest protestant. Just over half the boys were casual laborers (59%); 41% of girls reported being unemployed. The majority of boys (32%) reported only living with their mothers while the majority (27%) of girls were either living only with their mother (14%) or living with both parents (14%). Seven of the adolescents' partners were interviewed of which three were male and four female. The mean age was 23 years, and the majority (57%) had incomplete secondary education, with 29% having higher education (S1 Table). Most of the partners (57%) were business owners, while a minority were casual labourers (29%) or unemployed (14%). Nineteen parents/ caregivers were interviewed, of which 26% were male and 74% female. Only a third of parents had completed education beyond primary school and were either self-employed (32%) or casual laborers (32%). Of the stakeholders interviewed, 64% were males and 36% were females; the majority were engaged in community activities (54%) such as Community Health Volunteers (CHVs), village elders and pastors. The

mean age of the mentors interviewed was 31 years and all had either completed secondary school or received higher education; 35% of the mentors were working as CHVs (S1 Table).

## Perceived effectiveness of the program

Respondents were asked to provide their thoughts on the impact the intervention had on the adolescents and others exposed to it. Most of the parents reported seeing positive improvements in attitudes and behaviours of the adolescents who attended the program, including greater self-confidence as well as receptivity to parental suggestions or advice, which in turn led to improved parent-adolescent relationships.

> "*What I have liked a lot about this program is the fact that it gives the youths a chance to express themselves. That is a very important thing because it gives them a chance to start understanding themselves. . . and become people who have focus for their future lives. I have really liked that because, my daughter being one of them, it was not easy talking out or just reporting something that is disturbing her mind or something that she wants. She used to keep quiet for long. . . nowadays, she expresses herself much better*" (*Father in Nakuru*)

> "*She stopped taking bhangi (marijuana), stopped sniffing glue. She stopped*! *Now she even bathes as she never used to bathe.*" *(Mother in Nakuru)*

Many adolescents reported gaining skills in managing their own mental health with simple techniques taught in the sessions like breathing exercises, narrowing down bigger challenges into smaller ones, reflecting on the specific root of a problem and trying to resolve it, as well as identifying a trusted individual to share their challenges with. They reported feeling more empowered to also provide peer support to others who may be going through similar social challenges affecting their mental health.

> "*We have learnt about managing problems in mental health. How to manage problems, stress. In managing problems, we have learnt 'keep doing, get going'. . . and to stand up for yourself.*" *(Adolescent girl (18yrs) in Nairobi)*

> "*Right now if a friend of mine has a problem, I will be able to help based on what I have been taught in this program. Previously if a friend told me he had stress, I would not know how to help or manage him. You are forced to avoid him only to hear later that he committed suicide. So, it helps the society a lot.*" (*Adolescent boy (19yrs) in Nakuru*)

Mentors reported that some of the case studies they created to demonstrate the linkages between mental health and other components of the curriculum included associations between depression and anxiety and poverty, lack of money and unemployment, uncertainty and despair experienced through early pregnancy, peer influence to take drugs, and family conflict and broken relationships including the death of a loved one. During the interviews, the adolescent respondents were able to identify and describe these associations.

> "*Now the situations that are there, you might find maybe you gave birth at an early age, and maybe your parent is a single parent, and by giving birth the parent feels offended and they will throw words at you (scold) all the time, maybe beat your child, or abuse them.*" *(Adolescent girl (18yrs) in Nakuru)*

> "*When you are having issues with your mental health, the pattern of how you are doing your day-to-day things will change. Like personally. . .I am doing small time business so when*

*something is stressing me, I won't be able to wake up that early, go to Gikomba (market) and pick those clothes (to sell)." (Adolescent girl (19yrs) in Nairobi)*

Some of the mentors reported personally benefiting from the program through applying gained skills to learn how to acknowledge and manage their own personal mental health challenges. Respondents who had attended or implemented similar programs mentioned that unlike other programs, BTG was unique because it included mental health as a topic in the curriculum, it offered free one-on-one psychological help sessions, and it focused on out-of-school adolescents, who are considered a highly marginalized and difficult to reach group in the community.

*"What I liked about the BTG program is that it specifically focused on out-of-school [adolescents], so it was another learning level for us as mentors because you get there is a very big gap between those who were going to school and the teenagers who are not going to school. The gap was so big in terms of grasping the challenges of life. . . so I love it because they have that specific target and it also made us learn a lot"* (*Mentor in Nakuru*)

Although several benefits were reported as a result of the program, respondents said that more could be done to create awareness of the program through the community meetings that took place around every four months. Respondents said that parents and stakeholders were informed of these meetings through word of mouth. However, more intensive outreach activities for example, through carrying out door to door communication campaigns would have further increased awareness of the program in the community.

## Capacity within the community to address the mental health issues of vulnerable adolescents

Interviews were conducted with study participants to understand if and how the program was able to improve the capacity of the mentors to address the mental health issues of out-of-school adolescents. The interview questions focused on what worked and what improvements needed to be made in the training of the mentors and implementing the group and one-on-one sessions.

**Training of mentors.**   In addition to being trained on topics covering SRHR, mental health and life skills, most mentors reported gaining skills on how to build a good rapport with adolescents, how to identify adolescents who needed referral to formal health services, and how to maintain confidentiality. Moreover, they reported gaining skills on how to prepare for each session, which included carrying out a self-evaluation before each session to ensure that they were psychologically prepared, as well as checking that they had the necessary manuals/materials needed. The mentors reported that the training strategies used by the trainers were interactive and included role plays and daily recaps, which they said were extremely useful in helping them process information during what they described as quite intense sessions, especially on mental health, which they reported was a new topic area for the majority of them.

*"I learnt that before going to the session there is understanding yourself, like you understand when you are angry and after understanding yourself, you learn how to deal with the issues, even when you are in the process of facilitation in case there is something that has [upset] you. Because you understand yourself, you know how to deal with it without showing the mentees that you are [upset]." (Mentor in Nairobi)*

*"There is the issue of values which we looked at deeply because previously [in group sessions] you impose your value. But I learnt taking my values lightly because the one I am addressing has their own values that I need to give space to"* (Mentor in Nakuru)

*"I learnt about facilitation skills and good communication. There is communication then there is good communication. . .how to build that rapport with the mentees. . . because you could meet them today and tomorrow they don't come depending on how you talk to them."* (Mentor in Nakuru)

Most mentors reported preferring having physical rather than online training sessions mainly due to technical issues experienced including limited digital capacity or access to smart phones, or poor internet connectivity and audibility. Although the curriculum had been translated into the local dialect "Kiswahili", they suggested that further contextualization should have occurred to translate some of the more complex Kiswahili into the *local "Sheng" (Kiswahili slang)* for easier understanding. Many mentors also recommended more frequent recap sessions integrated into the training and also an extension of the training beyond the three-week period to allow for new information to be clearly understood. Furthermore, they requested the provision of training certificates that would allow them to utilise their newly gained skills beyond the BTG program.

*"We did learn a lot of skills, on how to handle the adolescents with mental health, but the only challenge was to translate this package into a language that they understand and that one really stressed us up a bit because you want to translate the whole content <<from>> Kiswahili . . . to Sheng." (Mentor in Nairobi)*

**Group sessions.** Mentors reported that the group sessions routinely started off with a morning prayer and a recap of the previous session followed by going through the topic of the day, a question and answer session, group discussions that included sharing of personal experiences, then games and fun activities. A few mentors reported going a step further and held celebrations such as birthdays during the sessions and invited speakers including professional counsellors and religious representatives to enhance understanding of certain topics. Many adolescents reported that they enjoyed the group sessions because they were interactive and they were able to actively participate; they also appreciated the openness and candidness of their peers and mentors. The adolescents felt that the sessions provided them with a platform where they were free to ask questions that they were not able to ask anyone else in their day to day lives, and the engagement enabled them to bond with other mentees.

*"Anytime I was down and I went to that group, I would leave feeling better because we had different kinds of people; we had those who would cheer you up, we had those who were just there. This gave me motivation for going there daily for I would learn something new and leave there happy." (Adolescent girl (19yrs) in Nairobi)*

*"We do more of skits, just to ensure that the point is actually sinking in, that they are understanding well, so as they act or perform, it helps to [internalize] the information that you are delivering" (Mentor in Nairobi)*

Both mentees and mentors provided recommendations on ways they thought the group sessions could be improved. The most common suggestions included more in-depth cover of topics on entrepreneurship, teen motherhood, and gender-based violence. They felt that these

were topics that significantly affect the adolescents in their community. Some mentees also requested more assurance and safeguards to ensure that what was being discussed in the meetings was kept confidential and not shared with others outside the sessions. Mentees also suggested that the venues selected could be improved by being near their homes and in quiet and not so congested surroundings that allow for better concentration and audibility. Several adolescents were very keen on having more mixed group sessions where opposite genders could learn more about each other, since only one session a month was a mixed gender group, where boys and girls would attend the same session. In addition, some adolescents had to attend the sessions with their babies however, it was reported that extra accommodations were needed to address the needs of the mothers and babies during the sessions.

"*There was limited mixing with the girls, and we also wanted to hear more about girls (from the girls), so if there can be some arrangements so that we have like at least two sessions a month together [with boys and girls], it would be better off.*" (*Adolescent boy (16yrs) in Nairobi*)

"*The other thing also that I was not happy about, though it's a blessing from God, many of those mentees have children, so you find that most of the time we have no place to put those children, like a day-care, to keep them there when we are busy with the girls. So, sometimes you are teaching the girls, children cry, there were so <many> distractions, so you find that their concentration becomes diverted towards their children.*" (*Mentor in Nairobi*)

Some mentees said that they would have liked the mentors to use pictures and videos for illustration and easier understanding of the topics. In addition, they recommended that regularly changing the format of how sessions were being delivered would create variety and excitement, for example having a few sessions outdoors or inviting mental health professionals like psychologists into the sessions from time to time.

**Enhanced delivery of a treatment focused PM+ through individual one-on-one sessions.** The majority of the adolescent respondents reported being aware of the one-on-one sessions describing them as sessions where the mentee was encouraged to privately go through the adapted PM+ intervention to address their specific personal challenges or issues identified by themselves or their mentors as causing significant distress in their day-to-day life. They reported having learned about the provision of this individualized treatment from the mentors during the group sessions, while a few reported having found out about them from fellow mentees who had already gone through the one-on-one sessions.

The majority of the respondents thought this individualized treatment allowed them to freely resolve personal issues like anger management in a safe space without the fear of unwanted disclosure as would be expected in a group setting. They also said they felt more connected to their mentors during this part of the intervention as the sessions gave them more opportunity to learn from their mentors who gave them more personalized attention without them being required to pay for the service, as would have been the case in the health care facilities and counseling centers. They reported that the sessions gave them more time to focus on how to manage their own problems as one had the mentor all to themselves as opposed to sharing them amongst 10 or more peers during the group sessions.

"*The mentor listened to me and encouraged me as a parent would. Sometimes the parent is not around and you need to discuss your problems with someone. However, you can't tell your friends because they will gossip about it. So, the mentor is the only one who can listen and advise you appropriately.*" (*Adolescent girl (17yrs) in Nakuru*

"*You know having the one-on-one sessions was like an advantage, you know out there going for counselling comes at a cost, so it was like we had been brought for free service. So it was something that we had to take advantage of and especially for me I really appreciated it because getting such a chance is very difficult.*" *(Adolescent boy (19yrs) in Nairobi)*

Despite the benefits of the one-on-one sessions, a minority of adolescents felt that the mentors should reconsider the best time to hold sessions. They said that some mentors held the one-on-one sessions at the end of the group sessions, meaning that mentees accessing the one-on-one sessions were left behind, exposing them to other group members as individuals who were having an issue. Some of the challenges mentors reported experiencing while providing the one-on-one sessions included reaching out to mentees who they felt needed help however the adolescent not being willing to open up and discussing their issues, experiencing personal challenges as a mentor that are emotionally draining and therefore finding it difficult to offer counselling, and also living in a densely populated environment and finding venues that offer privacy where sessions can be conducted confidentially. Mentors mentioned that some sessions had to be held at the adolescent's home which was not always a conducive environment.

Respondents who did not attend the one-on-one sessions reported that this was because they either had not yet faced an issue they were unable to handle by themselves or that they had already been equipped with sufficient skills to handle their current challenges during the PM+ group sessions, others reported not having enough free time to attend the sessions.

### Access to quality mental health information and services in the community

Respondents were asked whether they thought having the BTG program increased their access to quality mental health information and services in the community. Access included their ability to attend the group and one-on-one sessions and also follow through with any health-related referrals recommended by the mentor to adolescents who needed more specialized care that could be provided through qualified health care professionals.

**Attendance.**   Although the majority of adolescents reported being able to attend at least one group session, most said they were not able to attend all the sessions. The main reason provided by mentees for non-attendance was work and looking for work to make a living in order to meet basic responsibilities such as providing food, clothes and shelter for themselves, their partners and children. Other reported barriers to attendance included competing activities such as household chores, involvement in political campaigns and church activities, negative external influences such as alcohol and substance use, negative peer influence not to attend, and other factors including lack of motivation or incentives and illnesses. Mentors reported that inconsistent attendance led to adolescents having to carry out repeat sessions, and not being exposed to the full intervention. Both mentors and adolescents suggested providing incentives to improve attendance that may consist of linkage to job opportunities and activities that can equip them with further entrepreneurial skills. They reported that other programs tended to offer financial incentives, dry foods or snacks, sanitary pads and diapers provided during the sessions.

"*Give each person an orange to eat after class as they go home. Sitting at the session hungry and not being given anything at the end makes them lose the motivation to attend and they decide to stay home.*" (*Adolescent girl (17yrs) in Nakuru*)

Some mentors said that they had identified these additional needs for adolescents and took the initiative of linking the mentees to other available community services that provided

training on technical skills such as tailoring or hairdressing, as well as organizations that were supplying sanitary pads and diapers for children. Mentors reported that these linkages improved adolescents attendance at the sessions.

"*Yes, there has been a lot of support that we are trying to offer. There are other programs that are running and we are trying to bring them on board also in regard to economic empowerment. Wherever they are running those programs, they normally invite us as mentors to select those we believe really need that opportunity, so we refer them (adolescents). . ."* (*Mentor in Nakuru*)

**Health-related referrals.** Mentors reported referring adolescents for a variety of reasons including severe mental illness that the mentor was not able to manage, rape, problematic alcohol and substance use, or other medical reasons such as physical injury and signs of malnourishment. Most mentees who refrained from following up on referrals said this was because they could not afford the high fare costs required for public transport to get to the facility and back. Some mentees had also mentioned to mentors during the intervention that service costs such as being required to pay some amount for registration, opening of their hospital patient file, and for counselling services at the public health facilities they preferred, presented a barrier to their following up on referrals. Some mentees mentioned that they were afraid that a CHV might identify them when accessing a facility and become exposed because the CHVs were linked to the local health facilities and apart from carrying out outreach activities in the communities they live in, they also use the facility as their base. Some mentees in addition said they were afraid to discuss their issues with a healthcare professional they were not familiar with, and in a few cases reported that they were not sure that confidentiality would be maintained at the facility. To overcome some of these barriers to access, some mentors reported that they would refer mentees to other mentors who had professional training in counselling. Mentors reported that the referral process could be improved by having a "youth friendly" point person who could manage the referral processes including following up on the mentees to ensure that they are able to access the health facility and address any barriers to access they may face.

## Discussion

This is one of the few studies that examines how task shifting of skills on mental health promotion, prevention and treatment to lay youth members can be embedded into empowerment-focused group sessions for highly vulnerable out- of-school adolescents. It also highlights some of the challenges that may arise and need to be addressed to improve the intervention's outcomes. The findings show that the BTG program was able to help adolescents become more confident in speaking up about their problems, provide them with essential skills to manage and treat anxiety or mild depression, and increase their social network to help develop resilience and coping mechanisms to address everyday adversities. Mentors were able to gain new skills including adhering to the core principles of psychological intervention delivery, and providing a psychosocial intervention that was both prevention and treatment focused to the adolescents in the program, and these skills could be transferrable beyond the duration of the program. Mentors were also able to use their gained skills to manage their own mental health challenges.

The study findings however revealed certain limitations and possible ways in which they could be addressed to strengthen the program. First, at the adolescent level, improved

engagement and retention of adolescents could be achieved through locating sessions in venues that are quiet and providing sessions that are engaging. Informal settlements generally are characterised by high density populations, and in such settings, it would be important for program implementors to support mentors in planning sessions that can be held during quieter times of the day. Sessions should be planned in advance and incorporate games and fun activities, to try and make them less monotonous. Furthermore, separate sessions can be arranged for mothers with babies with provisions for child-friendly play activities to keep babies entertained and allow mothers time to focus during the sessions. There will be adolescents who may not be able to attend all the sessions mainly due to looking for work or having to go to work to sustain their livelihoods. Mentors should be made aware of this possibility during the trainings and be ready to address this situation by being flexible and creative about how they continuously administer the curriculum to the adolescents. Some possible suggestions include repeating sessions or even having audio recordings of each topic that can be shared with adolescents at the end of each session, to listen to in their free time. It would be important for mentors to discuss what would work best for the adolescents in their group sessions.

This study demonstrated that the attendance of adolescents whose immediate daily needs have not been met is likely to be low. Therefore, throughout the implementation of such a program, it would be important to carry out informal discussions with beneficiaries to understand what immediate needs are not being met by the program, and work with them to see how they can be met. In this study, for example, adolescents were in need of sanitary pads, diapers for their children, and gaining skills to be able to generate income. Some of these needs can be met through already available services in the community, therefore it will be advisable for the mentors and program staff to carry out a mapping of available free services and products available within the community, for example provided by government, non-governmental, and faith-based organizations. Mentors can then use this information to link up adolescents to these providers.

At the mentor level, the study revealed that mentors who have never been exposed to facilitating mental health sessions may require more time to understand and absorb the mental health content in the training materials. It is therefore important to understand the level of experience of mentors in delivering such content and tailoring the training accordingly. Possible alterations include carrying out routine assessments and using this to inform if further training is required, and carrying out routine refresher trainings. Furthermore, the findings highlighted the importance of ensuring that the mental health of the mentors is being taken care of throughout the program. Some mentors reported that there were times they felt overwhelmed with their own life challenges and were therefore not in the right mindset to deliver the psychosocial sessions. One way of addressing this is ensuring that prior to being enrolled into the program, mentors should undergo a screening by a clinical psychologist to assess their mental health status and use this to inform whether it would be suitable for them to participate in such a program. Throughout the duration of such a program, routine check-ins with the mentors should take place to monitor and address their well-being. Finally, during the group and one on one sessions, the study also revealed that mentors may need to take a little more time initially to allow for trust to build between them and the mentees before engaging with the curriculum and addressing sensitive issues.

At the program level, prior to implementation of such a program, implementors need to ensure that the curriculum has been contextualised to the environment in which it is being implemented. For example, in this study, ensuring that the language was translated into local *"Sheng" (Kiswahili slang)* would have allowed for easier understanding. Also, although a referral system was set up, there were challenges in its implementation. Adolescents complained about not being able to access the public health facilities due to costs that are likely to be

incurred through this process. This includes transport and service costs, such as being required to pay some amount for registration, opening of their hospital patient file, and for counselling services however subsidized. Some adolescents who were able to access the facilities reported being scared that they would be identified by the health workers who resided within their community necessitating them to prefer facilities some distance away from their community and hence further increasing their transport cost needs. One potential way to overcome some of these challenges would be to identify any ongoing outreach/ mobile health services that are being offered that the adolescent can access closer to their home. In addition, mentors can try identifying non-governmental organisations providing relevant health services outside of the community that may be willing to send a counsellor once every couple of weeks to come and provide services in their community. Respondents could also benefit from having a "youth friendly" point person to manage the referral process.

The strengths of the study are that it provides different perspectives of the program from the adolescents, their partners, mentors, parents, and community stakeholders. The interviews also took place towards the end of the study, but before it had ended, which allowed for less recall bias and meant that respondents had experienced close to a year of the program and were able to provide deeper insights regarding their experiences. Some of the limitations of the study include that there were a small number of fathers and partners interviewed, which limited our understanding of their experiences of the program. The study would have also benefitted from interviewing the CO implementors and clinical psychologists on their experiences in designing and implementing the program.

At a policy level, the study provides evidence on how to actualise the WHO's recommendation and the Kenya Mental Health Policy (2015–2030) objective of integrating mental health and social care, including promotion and prevention, into already existing community-based services, in order to provide a more comprehensive approach to mental health services. Further research is required on understanding the effectiveness and cost-effectiveness of integrated programs on mental health outcomes.

## Supporting information

**S1 Table. Socio-demographics of interviewed participants.**
(DOCX)

**S2 Table. Description of the coding tree for the qualitative interviews.**
(DOCX)

## Acknowledgments

The authors would first like to thank all the adolescents, mentors, parents and stakeholders who participated in this study. Their involvement has helped us understand how we can best provide marginalised adolescents with holistic care that is tailored to their needs. We would also like to thank the implementing partners REPACTED and Integrated Education for Community Empowerment (IECE), as well as the clinical psychologist Dorcas Khasowa, for the high-quality program implementation. We thank all the Population Council staff involved in this study, including Dr Karen Austrian, one of the study co-investigators; Daniel Mwanga, Stephen Kizito and Eva Muluve who developed and maintained the various data platforms; Grace Wanjala and Florence Thungu and all the data collectors and study co-ordinators who carried out the data collection, management and analyses and Joyce Altman for copy editing the manuscript. Finally, we dedicate this study to the late Mercy Nzioki, the Population

Council-Kenya adolescent program manager who oversaw the integration of the curricula, training the mentors and provided high quality supervision to the implementation of the study.

## Author Contributions

**Conceptualization:** Beth Kangwana, Manasi Kumar.

**Data curation:** Beth Kangwana.

**Formal analysis:** Beth Kangwana.

**Methodology:** Beth Kangwana, Manasi Kumar.

**Project administration:** Joan Mutahi, Manasi Kumar.

**Supervision:** Beth Kangwana, Joan Mutahi, Manasi Kumar.

**Validation:** Beth Kangwana, Joan Mutahi, Manasi Kumar.

**Writing – original draft:** Beth Kangwana.

**Writing – review & editing:** Joan Mutahi, Manasi Kumar.

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
