## [Decision Letter · Decision Letter 0]

20 Sep 2023

PONE-D-23-25047Experiences of integrating a psychological intervention into a youth-led empowerment program targeting out-of-school adolescents, in urban informal settlements in Kenya: A qualitative studyPLOS ONE

Dear Dr. Kangwana,

Thank you for submitting your manuscript to PLOS ONE. After careful consideration, we feel that it has merit but does not fully meet PLOS ONE’s publication criteria as it currently stands. Therefore, we invite you to submit a revised version of the manuscript that addresses the points raised during the review process.

Please address the reviewers' comments regarding the rigor of the qualitative analytical approach used. 

We look forward to receiving your revised manuscript.

Kind regards,

Lea Sacca

Academic Editor

PLOS ONE

Journal Requirements:

4. Please ensure that you include a title page within your main document. We do appreciate that you have a title page document uploaded as a separate file, however, as per our author guidelines (http://journals.plos.org/plosone/s/submission-guidelines#loc-title-page) we do require this to be part of the manuscript file itself and not uploaded separately.

5. Please amend your manuscript to include your abstract after the title page.

Reviewers' comments:

Reviewer's Responses to Questions

**Comments to the Author**

1. Is the manuscript technically sound, and do the data support the conclusions?

Reviewer #1: Yes

Reviewer #2: Partly

2. Has the statistical analysis been performed appropriately and rigorously? 

Reviewer #1: I Don't Know

Reviewer #2: No

3. Have the authors made all data underlying the findings in their manuscript fully available?

Reviewer #1: Yes

Reviewer #2: Yes

4. Is the manuscript presented in an intelligible fashion and written in standard English?

Reviewer #1: Yes

Reviewer #2: Yes

5. Review Comments to the Author

Reviewer #1: I enjoyed reviewing your paper and found that your study promotes an alternative process to provide mental health assistance to at risk adolescents. This demonstration project may serve well as an example for other low and middle income countries on how to access this adolescent population. In fact, it could be model for high income countries.

Reviewer #2: The manuscript presents a narrative report on a multi-component program for integrating mental health concerns into community-based services for out-of-school youth in two areas in Kenya. The qualitative review of the program focuses on the use of the program by youth, their parents, and the mentors that provided the program content. The structure of the narrative provides selected responses to the varied aspects of the program by the youth, their parents, and the mentors. The narrative does provide insights into the reactions that these subjects had. However, that content is of limited value in determining if the program was generally positively received or effective. Both positive and negative summaries of the interviews conducted are presented in those paragraphs. The value of the manuscript, therefore, becomes quiet limited. Here are the positive elements of the manuscript: 1. The introduction is comprehensive and provides readers with a clear understanding of the importance of the issues being addressed by the program and the nature of program content. 2. The program content selected for delivery is very appropriate. 3. The description of the training used for program delivery mentors and the coders of interviews was clear. 4. The use of individual interviews and focus groups is a sound choice; 5. The interest in obtaining a balanced sample of participants for the individual interviews was also a good choice. Major problems with the manuscript are as follows: 1. The study and program enrolled a substantial number of youth. This is important, but in order to understand the true impact of the program, use and attendance statistics need to be provided. From the current content, one could conclude that less than 50 youth were involved with enough interest to complete interviews. 2. The interviews conducted were evaluated using a code book that is supplied, but no information about the codes or their categories is provided in the main body. This is an important gap that must be filled in order for the manuscript to make a significant contribution. 3. Using information from the coding system, the varied sections contained in the results portion should provide descriptions of positive and negative aspects of the program. Comments should then follow along with those reports. 4. Did the group gather any summary reports that reflected that youth and their parents found the program valuable in the form of a global rating? If they did, what percent of participants provided positive reports? 5. Without the specific information in items 3 and 4, the discussion section is problematic. 5a. The discussion provides a generally positive report without real data to support it. It appears to be written based upon reflections that the authors have conducted based upon their extensive knowledge of the program and participants reactions which is not shared with the reader of the manuscript. Without the needed details, readers have to trust that the authors are providing accurate information. That level of trust must be supported with data so that readers can determine if the conclusions are warranted. 6. The results section includes a number of ideas that should be included in the discussion section. These ideas primarily involve steps that could be taken to improve content and delivery. These ideas should be incorporated in the discussion section. 6a. it would be useful to have a possible division in the discussion section: program benefits and program limitations with suggestions on how to improve the program.

Overall, the study may have hidden value that does not come across in the current manuscript. The program, its delivery, and its review seem to have been conducted thoughtfully and carefully. The manuscript does not provide enough details and data about results to tell the story of a possibly useful endeavor that others may want to replicate.

6. PLOS authors have the option to publish the peer review history of their article (what does this mean?). If published, this will include your full peer review and any attached files.

Reviewer #1: No

Reviewer #2: **Yes: **Richard Gallagher, Ph.D.

---

## [Author Response · Author response to Decision Letter 0]

21 Dec 2023

To Whom it May Concern, 

 We want to thank the reviewers for taking time to review the manuscript and provide their comments which we believe we have now addressed. We provide replies to each of the comments raised in blue italics below. 

The manuscript should now meet PLOS ONE’s manuscript style requirements, including how the files are named. Please let us know if we may have missed anything. 

We believe we only entered financial information in the portal under the ‘financial information’ section and not in the manuscript (as directed by the submission guidelines). The grant was made to Population Council- Kenya and not an individual therefore we have amended that. Beyond this change we would like to request further guidance on where the mismatch lies. Thank you!

We have uploaded the de-identified datasets onto Harvard Dataverse. The data can be cited as follows:

Kangwana, Beth, 2023, "Experiences of integrating a psychological intervention into a youth-led empowerment program", https://doi.org/10.7910/DVN/W8NWWD, Harvard Dataverse, V1.

4. Please ensure that you include a title page within your main document. We do appreciate that you have a title page document uploaded as a separate file, however, as per our author guidelines (http://journals.plos.org/plosone/s/submission-guidelines#loc-title-page) we do require this to be part of the manuscript file itself and not uploaded separately.

The title page has now been included within the main document. 

5. Please amend your manuscript to include your abstract after the title page.

The abstract is now available after the title page. 

Reviewers' comments:

Reviewer's Responses to Questions

Comments to the Author

5. Review Comments to the Author

Reviewer #1: I enjoyed reviewing your paper and found that your study promotes an alternative process to provide mental health assistance to at risk adolescents. This demonstration project may serve well as an example for other low and middle income countries on how to access this adolescent population. In fact, it could be model for high income countries.

Reviewer #2: The manuscript presents a narrative report on a multi-component program for integrating mental health concerns into community-based services for out-of-school youth in two areas in Kenya. The qualitative review of the program focuses on the use of the program by youth, their parents, and the mentors that provided the program content. The structure of the narrative provides selected responses to the varied aspects of the program by the youth, their parents, and the mentors. The narrative does provide insights into the reactions that these subjects had. However, that content is of limited value in determining if the program was generally positively received or effective. Both positive and negative summaries of the interviews conducted are presented in those paragraphs. The value of the manuscript, therefore, becomes quiet limited. Here are the positive elements of the manuscript: 1. The introduction is comprehensive and provides readers with a clear understanding of the importance of the issues being addressed by the program and the nature of program content. 2. The program content selected for delivery is very appropriate. 3. The description of the training used for program delivery mentors and the coders of interviews was clear. 4. The use of individual interviews and focus groups is a sound choice; 5. The interest in obtaining a balanced sample of participants for the individual interviews was also a good choice. Major problems with the manuscript are as follows: 

1. The study and program enrolled a substantial number of youth. This is important, but in order to understand the true impact of the program, use and attendance statistics need to be provided. From the current content, one could conclude that less than 50 youth were involved with enough interest to complete interviews. 

We thank the reviewer for this comment. The aim of this qualitative study was not to understand the true impact of the program but rather to gain program insights into participants’ perspectives on 1) perceived changes in psychosocial outcomes, 2) the capacity of community health mentors to address mental health challenges of adolescents and 3) access to quality mental health information and services in the community. On re-reading the introduction we have decided to rephrase the specific objectives of the study by eliminating the phrase ‘understand the effect of the integration’ to ‘gain insights into participant’s perspectives on the program’, we believe the word ‘effect’ is misleading for a qualitative study (line 40-41). 

As indicated in the manuscript, the sample size for the qualitative interviews was based on the number of participants that could be interviewed given the financial budget available. The budget allowed us to carry out 85 interviews (44 adolescents, 7 partners, 19 parents and 11 stakeholders in-depth interviews, and four focus group discussions with 17 mentors) and, as indicated in the manuscript we felt that we were able to reach data saturation by the end of the interviews (line 152). We have added in a reference to support the justification for the sample size for this study. We do note in the manuscript that we replaced respondents who were not able to be interviewed with other respondents with similar characteristics (line 167-171). Replacements were rare for the adolescents but more common for the parents and stakeholders who had competing work and home related priorities and could not avail themselves during the period that the interviewers were in the field. 

The study was able to enroll just over 400 out-of-school adolescents and retain 339 of them by the endline survey. In parallel to this qualitative survey we carried out a quantitative survey where we were able to interview all 339 adolescents; some of the findings have been published in ‘Kumar M, Mutahi J, Kangwana B. Leave no one behind: integrated sexual and reproductive, mental health and psychosocial programming for not in employment, education or training (NEET) adolescents and youth in sub-Saharan Africa. BMJ Global Health 2023; 8:e013394.’ Since the quantitative study was a before-and-after study with no control arm, we felt that the level of evidence was not sufficient to draft a stand-alone manuscript and therefore incorporated the findings into a commentary. We hope that the findings from this feasibility study will help us refine the program and carry out a more robust quantitative study such as a cluster randomized controlled trial, in order to assess the program’s effectiveness in improving MH outcomes (line 592-593).

2. The interviews conducted were evaluated using a code book that is supplied, but no information about the codes or their categories is provided in the main body. This is an important gap that must be filled in order for the manuscript to make a significant contribution. 

We thank the author for providing this valuable feedback. We have now entered details of the codebook, including the codes in the methodology section, and referenced the codebook available in the supplement section as follows:

The guides were developed by the research team and reviewed by the project’s clinical psychologists and CBO implementors from both sites, who designed them to address the objectives of the qualitative survey and explore opinions and experiences of each component of the intervention (line 174-176). 

An initial starting list of codes was developed and included in a code-book based on the program’s theory of change and interview guides, and then additional codes were added as new themes emerged from the data (26) (S2 Table). In the end, codes were categorized into general knowledge and perceptions of the overall program; mentor training sessions, safe space group meetings, one one-on-one sessions and the referral process (line 212-217).

3. Using information from the coding system, the varied sections contained in the results portion should provide descriptions of positive and negative aspects of the program. Comments should then follow along with those reports. 

We believe we have now addressed this important point. The results section is ordered according to the codes in the codebook. Under each subheading (code) we have structured the text to first go through the reported positive aspects of the component followed by identified weaknesses and recommendations for improvements that were suggested by the respondents. We have made sure that any thoughts/ opinions that came from the authors have been moved to the discussion section. 

4. Did the group gather any summary reports that reflected that youth and their parents found the program valuable in the form of a global rating? If they did, what percent of participants provided positive reports? 

During the qualitative survey, we did not gather any summary reports that reflected whether participants found the program valuable. As discussed under comment 1, for this qualitative survey we were interested in capturing the varied experiences of the program. This data will help in informing how the program could be improved for better user experience.

 Also, as mentioned in comment 1, aside from the qualitative survey we did carry out a quantitative before and after study that was published in a commentary. In the quantitative survey, we measured the effect of the intervention on certain mental health outcomes. The quantitative findings showed that out of 339 adolescents followed up to endline, a total of 70% of the adolescents were able to freely access the individual psychological counseling sessions delivered by the community mentors. We observed improvements in mental health literacy scores of both mentors (mean difference of score +13.8) and mentees (mean difference of score +2.8). There were also improvements in the General Help Seeking behaviors suggesting that more adolescents reported that they would seek help from a MH professional or mentor if they had a personal or emotional problem. We did not want to draft a stand-alone manuscript on the quantitative study because we felt that not having a control group would make it difficult to know if the outcomes observed were as a result of the intervention or due to other factors outside of the intervention. Since this is the first time (we are aware) that such the integration of a MH component into community-based empowerment sessions has taken place and experiences documented, we are using these study findings to assess the feasibility of this integration taking place. We hope to use the qualitative findings to refine/ improve user experiences of the intervention. 

5. Without the specific information in items 3 and 4, the discussion section is problematic. 5a. The discussion provides a generally positive report without real data to support it. It appears to be written based upon reflections that the authors have conducted based upon their extensive knowledge of the program and participants reactions which is not shared with the reader of the manuscript. Without the needed details, readers have to trust that the authors are providing accurate information. That level of trust must be supported with data so that readers can determine if the conclusions are warranted.

We hope that the issues raised here by the author have now been adequately addressed under items 3 and 4 and 6.

6. The results section includes a number of ideas that should be included in the discussion section. These ideas primarily involve steps that could be taken to improve content and delivery. These ideas should be incorporated in the discussion section. 

As reported under comment 3, we have gone through the results section and included only the responses that were provided in the interviews, any other information has been moved to the methods section, describing the intervention, and to the discussion section. We have left the recommendations provided by the respondents in the results section. We have however also included the main recommendations made by the respondents in the discussion section, and have added in additional thoughts from the design and implementation team/ authors, to help address some of the findings that came up during the interviews. 

6a. it would be useful to have a possible division in the discussion section: program benefits and program limitations with suggestions on how to improve the program.

We thank the reviewer for this suggestion. We have re-worked the discussion section and organized it into the first section which discusses the program benefits. The second section discusses program challenges and provides suggestions on how they can be overcome. 

Overall, the study may have hidden value that does not come across in the current manuscript. The program, its delivery, and its review seem to have been conducted thoughtfully and carefully. The manuscript does not provide enough details and data about results to tell the story of a possibly useful endeavor that others may want to replicate.

We thank the reviewer for his valuable feedback and hope with the changes made to the manuscript, that the purpose of the manuscript is clearer.

Yours sincerely, 

Beth Kangwana

---

## [Decision Letter · Decision Letter 1]

28 Feb 2024

Experiences of integrating a psychological intervention into a youth-led empowerment program targeting out-of-school adolescents, in urban informal settlements in Kenya: A qualitative study

PONE-D-23-25047R1

Dear Dr. Kangwana,

We’re pleased to inform you that your manuscript has been judged scientifically suitable for publication and will be formally accepted for publication once it meets all outstanding technical requirements.

Kind regards,

Lea Sacca

Academic Editor

PLOS ONE

Additional Editor Comments (optional):

Reviewers' comments:

Reviewer's Responses to Questions

**Comments to the Author**

1. If the authors have adequately addressed your comments raised in a previous round of review and you feel that this manuscript is now acceptable for publication, you may indicate that here to bypass the “Comments to the Author” section, enter your conflict of interest statement in the “Confidential to Editor” section, and submit your "Accept" recommendation.

Reviewer #1: All comments have been addressed

2. Is the manuscript technically sound, and do the data support the conclusions?

Reviewer #1: Yes

3. Has the statistical analysis been performed appropriately and rigorously? 

Reviewer #1: N/A

4. Have the authors made all data underlying the findings in their manuscript fully available?

Reviewer #1: Yes

5. Is the manuscript presented in an intelligible fashion and written in standard English?

Reviewer #1: Yes

6. Review Comments to the Author

Reviewer #1: I am pleased with the manuscript on my second review and I look forward to seeing it published in the near future.

7. PLOS authors have the option to publish the peer review history of their article (what does this mean?). If published, this will include your full peer review and any attached files.

Reviewer #1: No

---

## [Editor Report · Acceptance letter]

25 Mar 2024

PONE-D-23-25047R1 

PLOS ONE

Dear Dr. Kangwana, 

I'm pleased to inform you that your manuscript has been deemed suitable for publication in PLOS ONE. Congratulations! Your manuscript is now being handed over to our production team.

Kind regards, 

on behalf of

Dr. Lea Sacca 

Academic Editor

PLOS ONE